# Local Decorrelation for Improved Pedestrian Detection

**Woonhyun Nam**[*]
StradVision, Inc.
woonhyun.nam@stradvision.com

**Piotr Dollár**
Microsoft Research
pdollar@microsoft.com

**Joon Hee Han**
POSTECH, Republic of Korea
joonhan@postech.ac.kr

## Abstract

Even with the advent of more sophisticated, data-hungry methods, boosted decision trees remain extraordinarily successful for fast rigid object detection, achieving top accuracy on numerous datasets. While effective, most boosted detectors use decision trees with orthogonal (single feature) splits, and the topology of the resulting decision boundary may not be well matched to the natural topology of the data. Given highly correlated data, decision trees with oblique (multiple feature) splits can be effective. Use of oblique splits, however, comes at considerable computational expense. Inspired by recent work on discriminative decorrelation of HOG features, we instead propose an efficient feature transform that removes correlations in local neighborhoods. The result is an overcomplete but locally decorrelated representation ideally suited for use with orthogonal decision trees. In fact, orthogonal trees with our locally decorrelated features outperform oblique trees trained over the original features at a fraction of the computational cost. The overall improvement in accuracy is dramatic: on the Caltech Pedestrian Dataset, we reduce false positives nearly tenfold over the previous state-of-the-art.

## 1 Introduction

In recent years object detectors have undergone an impressive transformation [11, 32, 14]. Nevertheless, boosted detectors remain extraordinarily successful for fast detection of quasi-rigid objects. Such detectors were first proposed by Viola and Jones in their landmark work on efficient sliding window detection that made face detection practical and commercially viable [35]. This initial architecture remains largely intact today: boosting [31, 12] is used to train and combine decision trees and a cascade is employed to allow for fast rejection of negative samples. Details, however, have evolved considerably; in particular, significant progress has been made on the feature representation [6, 9, 2] and cascade architecture [3, 8]. Recent boosted detectors [1, 7] achieve state-of-the-art accuracy on modern benchmarks [10, 22] while retaining computational efficiency.

While boosted detectors have evolved considerably over the past decade, decision trees with orthogonal (single feature) splits – also known as axis-aligned decision trees – remain popular and predominant. A possible explanation for the persistence of orthogonal splits is their efficiency: oblique (multiple feature) splits incur considerable computational cost during both training and detection. Nevertheless, oblique trees can hold considerable advantages. In particular, Menze et al. [23] recently demonstrated that oblique trees used in conjunction with random forests are quite effective given high dimensional data with heavily correlated features.

To achieve similar advantages while avoiding the computational expense of oblique trees, we instead take inspiration from recent work by Hariharan et al. [15] and propose to *decorrelate* features prior to applying *orthogonal* trees. To do so we introduce an efficient feature transform that removes correlations in *local* image neighborhoods (as opposed to decorrelating features globally as in [15]). The result is an overcomplete but *locally decorrelated* representation that is ideally suited for use with orthogonal trees. In fact, orthogonal trees with our locally decorrelated features require estimation of fewer parameters and actually outperform oblique trees trained over the original features.

---

[*]This research was performed while W.N. was a postdoctoral researcher at POSTECH.

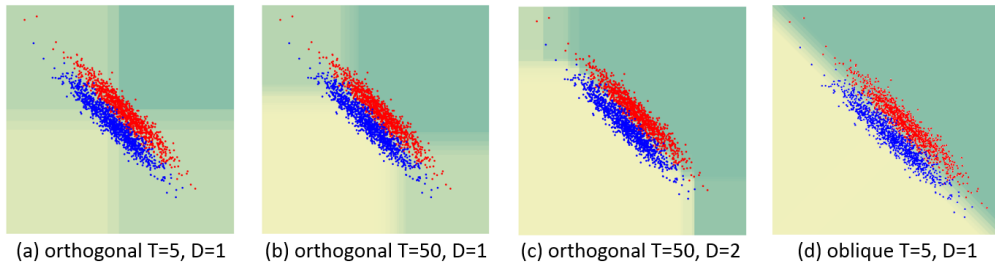

| (a) orthogonal T=5, D=1 | (b) orthogonal T=50, D=1 | (c) orthogonal T=50, D=2 | (d) oblique T=5, D=1 |

Figure 1: A comparison of boosting of orthogonal and oblique trees on highly correlated data while varying the number ($T$) and depth ($D$) of the trees. Observe that orthogonal trees generalize poorly as the topology of the decision boundary is not well aligned to the natural topology of the data.

We evaluate boosted decision tree learning with decorrelated features in the context of pedestrian detection. As our baseline we utilize the aggregated channel features (ACF) detector [7], a popular, top-performing detector for which source code is available online. Coupled with use of deeper trees and a denser sampling of the data, the improvement obtained using our *locally decorrelated channel features* (LDCF) is substantial. While in the past year the use of deep learning [25], motion features [27], and multi-resolution models [36] has brought down log-average miss rate (MR) to under $40\%$ on the Caltech Pedestrian Dataset [10], LDCF reduces MR to under $25\%$. This translates to a nearly tenfold reduction in false positives over the (very recent) state-of-the-art.

The paper is organized as follows. In §2 we review orthogonal and oblique trees and demonstrate that orthogonal trees trained on decorrelated data may be equally or more effective as oblique trees trained on the original data. We introduce the baseline in §3 and in §4 show that use of oblique trees improves results but at considerable computational expense. Next, in §5, we demonstrate that orthogonal trees trained with locally decorrelated features are efficient and effective. Experiments and results are presented in §6. We begin by briefly reviewing related work next.

## 1.1   Related Work

**Pedestrian Detection**: Recent work in pedestrian detection includes use of deformable part models and their extensions [11, 36, 26], convolutional nets and deep learning [33, 37, 25], and approaches that focus on optimization and learning [20, 18, 34]. Boosted detectors are also widely used. In particular, the channel features detectors [9, 1, 2, 7] are a family of conceptually straightforward and efficient detectors based on boosted decision trees computed over multiple feature channels such as color, gradient magnitude, gradient orientation and others. Current top results on the INRIA [6] and Caltech [10] Pedestrian Datasets include instances of the channel features detector with additional mid-level edge features [19] and motion features [27], respectively.

**Oblique Decision Trees**: Typically, decision trees are trained with orthogonal (single feature) splits; however, the extension to oblique (multiple feature) splits is fairly intuitive and well known, see e.g. [24]. In fact, Breiman's foundational work on random forests [5] experimented with oblique trees. Recently there has been renewed interest in random forests with oblique splits [23, 30] and Marin et al. [20] even applied such a technique to pedestrian detection. Likewise, while typically orthogonal trees are used with boosting [12], oblique trees can easily be used instead. The contribution of this work is not the straightforward coupling of oblique trees with boosting, rather, we propose a local decorrelation transform that eliminates the necessity of oblique splits altogether.

**Decorrelation**: Decorrelation is a common pre-processing step for classification [17, 15]. In recent work, Hariharan et al. [15] proposed an efficient scheme for estimating covariances between HOG features [6] with the goal of replacing linear SVMs with LDA and thus allowing for fast training. Hariharan et al. demonstrated that the *global* covariance matrix for a detection window can be estimated efficiently as the covariance between two features should depend only on their relative offset. Inspired by [15], we likewise exploit the stationarity of natural image statistics, but instead propose to estimate a *local* covariance matrix shared across all image patches. Next, rather than applying *global decorrelation*, which would be computationally prohibitive for sliding window detection with a nonlinear classifier[1], we instead propose to apply an efficient *local decorrelation* transform. The result is an overcomplete representation well suited for use with orthogonal trees.

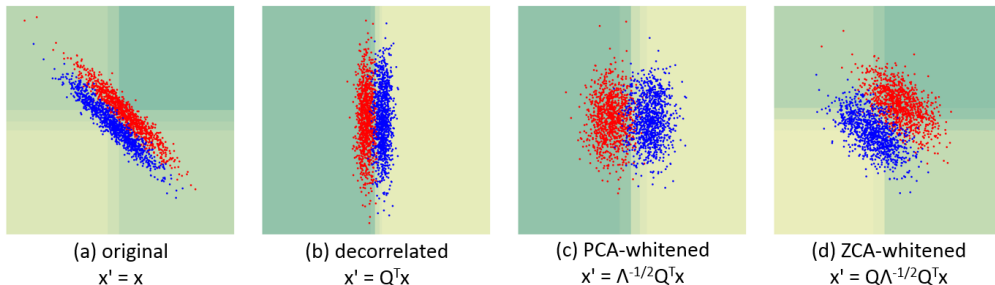

| (a) original | (b) decorrelated | (c) PCA-whitened | (d) ZCA-whitened |
|---|---|---|---|
| x' = x | x' = Q$^\mathsf{T}$x | x' = $\Lambda^{-1/2}$Q$^\mathsf{T}$x | x' = Q$\Lambda^{-1/2}$Q$^\mathsf{T}$x |

Figure 2: A comparison of boosting with orthogonal decision trees ($T = 5$) on transformed data. Orthogonal trees with both decorrelated and PCA-whitened features show improved generalization while ZCA-whitening is ineffective. *Decorrelating* the features is critical, while *scaling* is not.

## 2 Boosted Decision Trees with Correlated Data

Boosting is a simple yet powerful tool for classification and can model complex non-linear functions [31, 12]. The general idea is to train and combine a number of weak learners into a more powerful strong classifier. Decision trees are frequently used as the weak learner in conjunction with boosting, and in particular orthogonal decision trees, that is trees in which every split is a threshold on a single feature, are especially popular due to their speed and simplicity [35, 7, 1].

The representational power obtained by boosting orthogonal trees is not limited by use of orthogonal splits; however, the number and depth of the trees necessary to fit the data may be large. This can lead to complex decision boundaries and poor generalization, especially given highly correlated features. Figure 1(a)-(c) shows the result of boosted orthogonal trees on correlated data. Observe that the orthogonal trees generalize poorly even as we vary the number and depth of the trees.

Decision trees with oblique splits can more effectively model data with correlated features as the topology of the resulting classifier can better match the natural topology of the data [23]. In oblique trees, every split is based on a linear projection of the data $z = \mathbf{w}^\mathsf{T}\mathbf{x}$ followed by thresholding. The projection $\mathbf{w}$ can be sparse (and orthogonal splits are a special case with $\|\mathbf{w}\|_0 = 1$). While in principle numerous approaches can be used to obtain $\mathbf{w}$, in practice linear discriminant analysis (LDA) is a natural choice for obtaining discriminative splits efficiently [16]. LDA aims to minimize within-class scatter while maximizing between-class scatter. $\mathbf{w}$ is computed from class-conditional mean vectors $\boldsymbol{\mu}_+$ and $\boldsymbol{\mu}_-$ and a class-independent covariance matrix $\boldsymbol{\Sigma}$ as follows:

$$\mathbf{w} = \boldsymbol{\Sigma}^{-1}(\boldsymbol{\mu}_+ - \boldsymbol{\mu}_-). \tag{1}$$

The covariance may be degenerate if the amount or underlying dimension of the data is low; in this case LDA can be regularized by using $(1 - \epsilon)\boldsymbol{\Sigma} + \epsilon\boldsymbol{I}$ in place of $\boldsymbol{\Sigma}$. In Figure 1(d) we apply boosted oblique trees trained with LDA on the same data as before. Observe the resulting decision boundary better matches the underlying data distribution and shows improved generalization.

The connection between whitening and LDA is well known [15]. Specifically, LDA simplifies to a trivial classification rule on whitened data (data whose covariance is the identity). Let $\boldsymbol{\Sigma} = \mathbf{Q}\boldsymbol{\Lambda}\mathbf{Q}^\mathsf{T}$ be the eigendecomposition of $\boldsymbol{\Sigma}$ where $\mathbf{Q}$ is an orthogonal matrix and $\boldsymbol{\Lambda}$ is a diagonal matrix of eigenvalues. $\mathbf{W} = \mathbf{Q}\boldsymbol{\Lambda}^{-\frac{1}{2}}\mathbf{Q}^\mathsf{T} = \boldsymbol{\Sigma}^{-\frac{1}{2}}$ is known as a *whitening* matrix because the covariance of $\mathbf{x}' = \mathbf{W}\mathbf{x}$ is the identity matrix. Given whitened data and means, LDA can be interpreted as learning the trivial projection $\mathbf{w}' = \boldsymbol{\mu}'_+ - \boldsymbol{\mu}'_- = \mathbf{W}\boldsymbol{\mu}_+ - \mathbf{W}\boldsymbol{\mu}_-$ since $\mathbf{w}'^\mathsf{T}\mathbf{x}' = \mathbf{w}'^\mathsf{T}\mathbf{W}\mathbf{x} = \mathbf{w}^\mathsf{T}\mathbf{x}$. Can whitening or a related transform likewise simplify learning of boosted decision trees?

Using standard terminology [17], we define the following related transforms: *decorrelation* ($\mathbf{Q}^\mathsf{T}$), *PCA-whitening* ($\boldsymbol{\Lambda}^{-\frac{1}{2}}\mathbf{Q}^\mathsf{T}$), and *ZCA-whitening* ($\mathbf{Q}\boldsymbol{\Lambda}^{-\frac{1}{2}}\mathbf{Q}^\mathsf{T}$). Figure 2 shows the result of boosting *orthogonal* trees on the variously transformed features, using the same data as before. Observe that with decorrelated and PCA-whitened features orthogonal trees show improved generalization. In fact, as each split is invariant to scaling of individual features, orthogonal trees with PCA-whitened and decorrelated features give identical results. *Decorrelating* the features is critical, while *scaling* is not. The intuition is clear: each split operates on a single feature, which is most effective if the features are decorrelated. Interestingly, the standard ZCA-whitened transform used by LDA is ineffective: while the resulting features are not technically correlated, due to the additional rotation by $\mathbf{Q}$ each resulting feature is a linear combination of features obtained by PCA-whitening.

# 3 Baseline Detector (ACF)

We next briefly review our baseline detector and evaluation benchmark. This will allow us to apply the ideas from §2 to object detection in subsequent sections. In this work we utilize the channel features detectors [9, 7, 1, 2], a family of conceptually straightforward and efficient detectors for which variants have been utilized for diverse tasks such as pedestrian detection [10], sign recognition [22] and edge detection [19]. Specifically, for our experiments we focus on pedestrian detection and employ the aggregate channel features (ACF) variant [7] for which code is available online[2].

Given an input image, ACF computes several feature channels, where each channel is a per-pixel feature map such that output pixels are computed from corresponding patches of input pixels (thus preserving image layout). We use the same channels as [7]: normalized gradient magnitude (1 channel), histogram of oriented gradients (6 channels), and LUV color channels (3 channels), for a total of 10 channels. We downsample the channels by 2x and features are single pixel lookups in the aggregated channels. Thus, given a $h \times w$ detection window, there are $h/2 \cdot w/2 \cdot 10$ candidate features (channel pixel lookups). We use RealBoost [12] with multiple rounds of bootstrapping to train and combine 2048 depth-3 decision trees over these features to distinguish object from background. Soft-cascades [3] and an efficient multiscale sliding-window approach are employed. Our baseline uses slightly altered parameters from [7] (RealBoost, deeper trees, and less downsampling); this increases model capacity and benefits our final approach as we report in detail in §6.

Current practice is to use the INRIA Pedestrian Dataset [6] for parameter tuning, with the test set serving as a validation set, see e.g. [20, 2, 9]. We utilize this dataset in much the same way and report full results on the more challenging Caltech Pedestrian Dataset [10]. Following the methodology of [10], we summarize performance using the *log-average miss rate* (MR) between $10^{-2}$ and $10^{0}$ false positives per image. We repeat all experiments 10 times and report the mean MR and standard error for every result. Due to the use of a log-log scale, even small improvements in (log-average) MR correspond to large reductions in false-positives. On INRIA, our (slightly modified) baseline version of ACF scores at $17.3\%$ MR compared to $17.0\%$ MR for the model reported in [7].

# 4 Detection with Oblique Splits (ACF-LDA)

In this section we modify the ACF detector to enable oblique splits and report the resulting gains. Recall that given input $\mathbf{x}$, at each split of an oblique decision tree we need to compute $z = \mathbf{w}^\intercal \mathbf{x}$ for some projection $\mathbf{w}$ and threshold the result. For our baseline pedestrian detector, we use $128 \times 64$ windows where each window is represented by a feature vector $\mathbf{x}$ of size $128/2 \cdot 64/2 \cdot 10 = 20480$ (see §3). Given the high dimensionality of the input $\mathbf{x}$ coupled with the use of thousands of trees in a typical boosted classifier, for efficiency $\mathbf{w}$ must be sparse.

**Local $\mathbf{w}$**: We opt to use $\mathbf{w}$'s that correspond to local $m \times m$ blocks of pixels. In other words, we treat $\mathbf{x}$ as a $h/2 \times w/2 \times 10$ tensor and allow $\mathbf{w}$ to operate over any $m \times m \times 1$ patch in a single channel of $\mathbf{x}$. Doing so holds multiple advantages. Most importantly, each pixel has strongest correlations to spatially nearby pixels [15]. Since oblique splits are expected to help most when features are strongly correlated, operating over local neighborhoods is a natural choice. In addition, using local $\mathbf{w}$ allows for faster lookups due to the locality of adjacent pixels in memory.

**Complexity**: First, let us consider the complexity of training the oblique splits. Let $d = h/2 \cdot w/2$ be the window size of a single channel. The number of patches per channel in $\mathbf{x}$ is about $d$, thus naively training a single split means applying LDA $d$ times – once per patch – and keeping $\mathbf{w}$ with lowest error. Instead of computing $d$ independent matrices $\mathbf{\Sigma}$ per channel, for efficiency, we compute $\overline{\mathbf{\Sigma}}$, a $d \times d$ covariance matrix for the entire window, and reconstruct individual $m^2 \times m^2$ $\mathbf{\Sigma}$'s by fetching appropriate entries from $\overline{\mathbf{\Sigma}}$. A similar trick can be used for the $\boldsymbol{\mu}$'s. Computing $\overline{\mathbf{\Sigma}}$ is $O(nd^2)$ given $n$ training examples (and could be made faster by omitting unnecessary elements). Inverting each $\mathbf{\Sigma}$, the bottleneck of computing Eq. (1), is $O(dm^6)$ but independent of $n$ and thus fairly small as $n \gg m$. Finally computing $z = \mathbf{w}^\intercal \mathbf{x}$ over all $n$ training examples and $d$ projections is $O(ndm^2)$. Given the high complexity of each step, a naive brute-force approach for training is infeasible.

**Speedup**: While the weights over training examples change at every boosting iteration and after every tree split, in practice we find it is unnecessary to recompute the projections that frequently. Table 1, rows 2-4, shows the results of ACF with oblique splits, updated every $T$ boosting iterations

|           | Shared $\boldsymbol{\Sigma}$ | $T$ | Miss Rate | Training |
|-----------|:---:|:---:|:---:|---:|
| ACF | - | - | $17.3 \pm .33$ | 4.93m |
| ACF-LDA-4 | No | 4 | $14.9 \pm .37$ | 303.57m |
| ACF-LDA-16 | No | 16 | $15.1 \pm .28$ | 78.11m |
| ACF-LDA-$\infty$ | No | $\infty$ | $17.0 \pm .22$ | 5.82m |
| ACF-LDA$^*$-4 | Yes | 4 | $14.7 \pm .29$ | 194.26m |
| ACF-LDA$^*$-16 | Yes | 16 | $15.1 \pm .12$ | 51.19m |
| ACF-LDA$^*$-$\infty$ | Yes | $\infty$ | $16.4 \pm .17$ | 5.79m |
| LDCF | Yes | - | $13.7 \pm .15$ | 6.04m |

Table 1: A comparison of boosted trees with orthogonal and oblique splits.

(denoted by ACF-LDA-$T$). While more frequent updates improve accuracy, ACF-LDA-16 has negligibly higher MR than ACF-LDA-4 but a nearly fourfold reduction in training time (timed using 12 cores). Training the brute force version of ACF-LDA, updated at every iteration and each tree split (7 interior nodes per depth-3 tree) would have taken about $5 \cdot 4 \cdot 7 = 140$ hours. For these results we used regularization of $\epsilon = .1$ and patch size of $m = 5$ (effect of varying $m$ is explored in §6).

**Shared $\boldsymbol{\Sigma}$**: The crux and computational bottleneck of ACF-LDA is the computation and application of a separate covariance $\boldsymbol{\Sigma}$ at each local neighborhood. In recent work on training linear object detectors using LDA, Hariharan et al. [15] exploited the observation that the statistics of natural images are translationally invariant and therefore the covariance between two features should depend only on their *relative* offset. Furthermore, as positives are rare, [15] showed that the covariances can be precomputed using natural images. Inspired by these observations, *we propose to use a single, fixed covariance $\boldsymbol{\Sigma}$ shared across all local image neighborhoods*. We precompute one $\boldsymbol{\Sigma}$ per channel and do not allow it to vary spatially or with boosting iteration. Table 1, rows 5-7, shows the results of ACF with oblique splits using fixed $\boldsymbol{\Sigma}$, denoted by ACF-LDA$^*$. As before, the $\boldsymbol{\mu}$'s and resulting $\mathbf{w}$ are updated every $T$ iterations. As expected, training time is reduced relative to ACF-LDA. Surprisingly, however, accuracy improves as well, presumably due to the implicit regularization effect of using a fixed $\boldsymbol{\Sigma}$. This is a powerful result we will exploit further.

**Summary**: ACF with local oblique splits and a single shared $\boldsymbol{\Sigma}$ (ACF-LDA$^*$-4) achieves $14.7\%$ MR compared to $17.3\%$ MR for ACF with orthogonal splits. The $2.6\%$ improvement in log-average MR corresponds to a nearly *twofold* reduction in false positives but comes at considerable computational cost. In the next section, we propose an alternative, more efficient approach for exploiting the use of a single *shared* $\boldsymbol{\Sigma}$ capturing correlations in *local* neighborhoods.

## 5 Locally Decorrelated Channel Features (LDCF)

We now have all the necessary ingredients to introduce our approach. We have made the following observations: (1) oblique splits learned with LDA over local $m \times m$ patches improve results over orthogonal splits, (2) a single covariance matrix $\boldsymbol{\Sigma}$ can be shared across all patches per channel, and (3) orthogonal trees with decorrelated features can potentially be used in place of oblique trees. This suggests the following approach: for every $m \times m$ patch $\mathbf{p}$ in $\mathbf{x}$, we can create a decorrelated representation by computing $\mathbf{Q}^\intercal\mathbf{p}$, where $\mathbf{Q}\boldsymbol{\Lambda}\mathbf{Q}^\intercal$ is the eigendecomposition of $\boldsymbol{\Sigma}$ as before, followed by use of orthogonal trees. However, such an approach is computationally expensive.

First, due to use of overlapping patches, computing $\mathbf{Q}^\intercal\mathbf{p}$ for every overlapping patch results in an overcomplete representation with a factor $m^2$ increase in feature dimensionality. To reduce dimensionality, we only utilize the top $k$ eigenvectors in $\mathbf{Q}$, resulting in $k < m^2$ features per pixel. The intuition is that the top eigenvectors capture the salient neighborhood structure. Our experiments in §6 confirm this: using as few as $k = 4$ eigenvectors per channel for patches of size $m = 5$ is sufficient. As our second speedup, we observe that the projection $\mathbf{Q}^\intercal\mathbf{p}$ can be computed by a series of $k$ convolutions between a channel image and each $m \times m$ filter reshaped from its corresponding eigenvector (column of $\mathbf{Q}$). This is possible because the covariance matrix $\boldsymbol{\Sigma}$ is shared across all patches per channel and hence the derived $\mathbf{Q}$ is likewise spatially invariant. Decorrelating all 10 channels in an entire feature pyramid for a $640 \times 480$ image takes about .5 seconds.

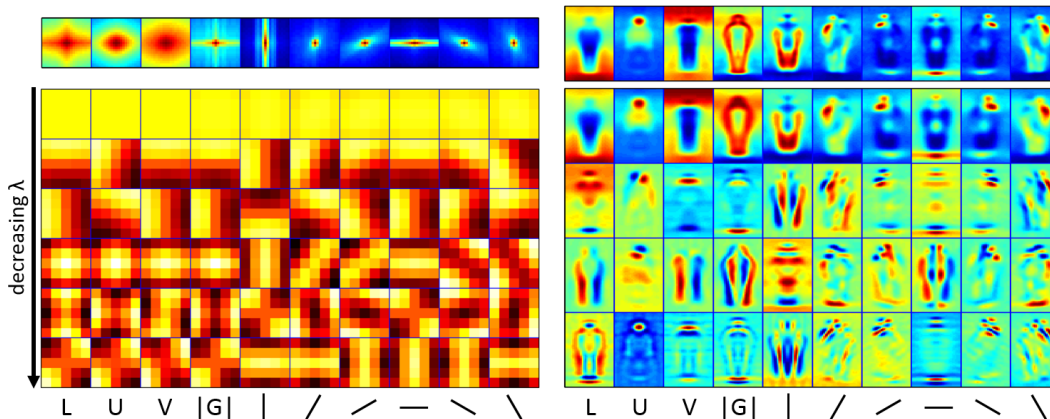

Figure 3: Top-left: autocorrelation for each channel. Bottom-left: learned decorrelation filters. Right: visualization of original and decorrelated channels averaged over positive training examples.

In summary, we modify ACF by taking the original 10 channels and applying $k = 4$ decorrelating (linear) filters per channel. The result is a set of 40 *locally decorrelated channel features* (LDCF). To further increase efficiency, we downsample the decorrelated channels by a factor of 2x which has negligible impact on accuracy but reduces feature dimension to the original value. Given the new locally decorrelated channels, all other steps of ACF training and testing are identical. The extra implementation effort is likewise minimal: given the decorrelation filters, a few lines of code suffice to convert ACF into LDCF. To further improve clarity, all source code for LDCF will be released.

Results of the LDCF detector on the INRIA dataset are given in the last row of Table 1. The LCDF detector (which uses orthogonal splits) improves accuracy over ACF with oblique splits by an additional 1% MR. Training time is significantly faster, and indeed, is only ~1 minute longer than for the original ACF detector. More detailed experiments and results are reported in §6. We conclude by (1) describing the estimation of $\Sigma$ for each channel, (2) showing various visualizations, and (3) discussing the filters themselves and connections to known filters.

**Estimating $\Sigma$**: We can estimate a spatially constant $\Sigma$ for each channel using any large collection of natural images. $\Sigma$ for each channel is represented by a spatial autocorrelation function $\Sigma_{(x,y),(x+\Delta x,y+\Delta y)} = C(\Delta x, \Delta y)$. Given a collection of natural images, we first estimate a separate autocorrelation function for each image and then average the results. Naive computation of the final function is $O(np^2)$ but the Wiener-Khinchin theorem reduces the complexity to $O(np \log p)$ via the FFT [4], where $n$ and $p$ denote the number of images and pixels per image, respectively.

**Visualization**: Fig. 3, top-left, illustrates the estimated autocorrelations for each channel. Nearby features are highly correlated and oriented gradients are spatially correlated along their orientation due to curvilinear continuity [15]. Fig. 3, bottom-left, shows the decorrelation filters for each channel obtained by reshaping the largest eigenvectors of $\Sigma$. The largest eigenvectors are smoothing filters while the smaller ones resemble increasingly higher-frequency filters. The corresponding eigenvalues decay rapidly and in practice we use the top $k = 4$ filters. Observe that the decorrelation filters for oriented gradients are aligned to their orientation. Finally, Fig. 3, right, shows original and decorrelated channels averaged over positive training examples.

**Discussion**: Our decorrelation filters are closely related to sinusoidal, DCT basis, and Gaussian derivative filters. Spatial interactions in natural images are often well-described by Markov models [13] and first-order stationary Markov processes are known to have sinusoidal KLT bases [29]. In particular, for the LUV color channels, our filters are similar to the discrete cosine transform (DCT) bases that are often used to approximate the KLT. For oriented gradients, however, the decorrelation filters are no longer well modeled by the DCT bases (note also that our filters are applied densely whereas the DCT typically uses block processing). Alternatively, we can interpret our filters as Gaussian derivative filters. Assume that the autocorrelation is modeled by a squared-exponential function $C(\Delta x) = \exp(-\Delta x^2/2l^2)$, which is fairly reasonable given the estimation results in Fig. 3. In 1D, the $k^{\text{th}}$ largest eigenfunction of such an autocorrelation function is a $k - 1$ order Gaussian derivative filter [28]. It is straightforward to extend the result to an anisotropic multivariate case in which case the eigenfunctions are Gaussian directional derivative filters similar to our filters.

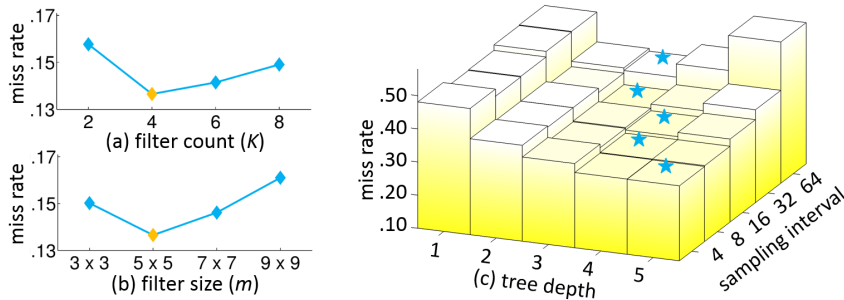

Figure 4: (a-b) Use of $k = 4$ local decorrelation filters of size $m = 5$ gives optimal performance. (c) Increasing tree depth while simultaneously enlarging the quantity of data available for training can have a large impact on accuracy (blue stars indicate optimal depth at each sampling interval).

|  | description | # channels | miss rate |
|---|---|---|---|
| 1. ACF | (modified) baseline | 10 | $17.3 \pm .33$ |
| 2. LDCF small $\lambda$ | decorrelation w $k$ smallest filters | $10k$ | $61.7 \pm .28$ |
| 3. LDCF random | filtering w $k$ random filters | $10k$ | $15.6 \pm .26$ |
| 4. LDCF LUV only | decorrelation of LUV channels only | $3k + 7$ | $16.2 \pm .37$ |
| 5. LDCF grad only | decorrelation of grad channels only | $3 + 7k$ | $14.9 \pm .29$ |
| 6. LDCF constant | decorrelation w constant filters | $10k$ | $14.2 \pm .34$ |
| 7. **LDCF** | proposed approach | $10k$ | $13.7 \pm .15$ |

Table 2: Locally decorrelated channels compared to alternate filtering strategies. See text.

## 6   Experiments

In this section, we demonstrate the effectiveness of locally decorrelated channel features (LDCF) in the context of pedestrian detection. We: (1) study the effect of parameter settings, (2) test variations of our approach, and finally (3) compare our results with the state-of-the-art.

**Parameters**: LDCF has two parameters: the count and size of the decorrelation filters. Fig. 4(a) and (b) show the results of LDCF on the INRIA dataset while varying the filter count ($k$) and size ($m$), respectively. Use of $k = 4$ decorrelation filters of size $m = 5$ improves performance up to ~4% MR compared to ACF. Inclusion of additional higher-frequency filters or use of larger filters can cause performance degradation. For all remaining experiments we fix $k = 4$ and $m = 5$.

**Variations**: We test variants of LDCF and report results on INRIA in Table 2. LDCF (row 7) outperforms all variants, including the baseline (1). Filtering the channels with the smallest $k$ eigenvectors (2) or $k$ random filters (3) performs worse. Local decorrelation of only the color channels (4) or only the gradient channels (5) is inferior to decorrelation of all channels. Finally, we test constant decorrelation filters obtained from the intensity channel L that resemble the first $k$ DCT basis filters. Use of unique filters per channel outperforms use of constant filters across all channels (6).

**Model Capacity**: Use of locally decorrelated features implicitly allows for richer, more effective splitting functions, increasing modeling capacity and generalization ability. Inspired by their success, we explore additional strategies for augmenting model capacity. For the following experiments, we rely solely on the training set of the Caltech Pedestrian Dataset [10]. Of the 71 minute long training videos (~128k images), we use every fourth video as validation data and the rest for training. On the validation set, LDCF outperforms ACF by a considerable margin, reducing MR from $46.2\%$ to $41.7\%$. We first augment model capacity by increasing the number of trees twofold (to 4096) and the sampled negatives fivefold (to 50k). Surprisingly, doing so reduces MR by an additional 4%. Next, we experiment with increasing maximum tree depth while simultaneously enlarging the amount of data available for training. Typically, every $30^{\text{th}}$ image in the Caltech dataset is used for training and testing. Instead, Figure 4(c) shows validation performance of LDCF with different tree depths while varying the training data sampling interval. The impact of maximum depth on performance is quite large. At a dense sampling interval of every $4^{\text{th}}$ frame, use of depth-5 trees (up from depth-2 for the original approach) improves performance by an additional $5\%$ to $32.6\%$ MR. Note that consistent with the generalization bounds of boosting [31], use of deeper trees requires more data.

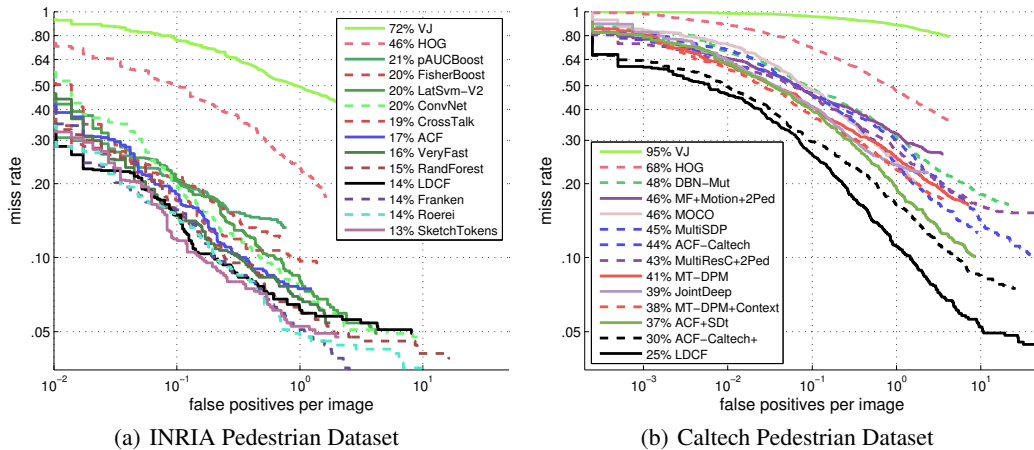

(a) INRIA Pedestrian Dataset　　　　(b) Caltech Pedestrian Dataset

Figure 5: A comparison of our LDCF detector with state-of-the-art pedestrian detectors.

**INRIA Results**: In Figure 5(a) we compare LDCF with state-of-the-art detectors on INRIA [6] using benchmark code maintained by [10]. Since the INRIA dataset is oft-used as a validation set, including in this work, we include these results for completeness only. LDCF is essentially tied for second place with Roerei [2] and Franken [21] and outperformed by ~1% MR by SketchTokens [19]. These approaches all belong to the family of channel features detectors, and as the improvements proposed in this work are orthogonal, the methods could potentially be combined.

**Caltech Results**: We present our main result on the Caltech Pedestrian Dataset [10], see Fig. 5(b), generated using the official evaluation code available online[3]. The Caltech dataset has become the standard for evaluating pedestrian detectors and the latest methods based on deep learning (Joint-Deep) [25], multi-resolution models (MT-DPM) [36] and motion features (ACF+SDt) [27] achieve under 40% log-average MR. For a complete comparison, we first present results for an augmented capacity ACF model which uses more (4096) and deeper (depth-5) trees trained with RealBoost using dense sampling of the training data (every 4[th] image). See preceding note on *model capacity* for details and motivation. This augmented model (ACF-Caltech+) achieves 29.8% MR, a considerable nearly 10% MR gain over previous methods, including the baseline version of ACF (ACF-Caltech). With identical parameters, locally decorrelated channel features (LDCF) further reduce error to 24.9% MR with substantial gains at higher recall. Overall, this is a massive improvement and represents a nearly 10x reduction in false positives over the previous state-of-the-art.

## 7  Conclusion

In this work we have presented a simple, principled approach for improving boosted object detectors. Our core observation was that effective but expensive oblique splits in decision trees can be replaced by orthogonal splits over *locally decorrelated* data. Moreover, due to the stationary statistics of image features, the local decorrelation can be performed efficiently via convolution with a fixed filter bank precomputed from natural images. Our approach is general, simple and fast.

Our method showed dramatic improvement over previous state-of-the-art. While some of the gain was from increasing model capacity, use of local decorrelation gave a clear and significant boost. Overall, we reduced false-positives tenfold on Caltech. Such large gains are fairly rare.

In the present work we did not decorrelate features across channels (decorrelation was applied independently per channel). This is a clear future direction. Testing local decorrelation in the context of other classifiers (e.g. convolutional nets or linear classifiers as in [15]) would also be interesting.

While the proposed locally decorrelated channel features (LDCF) require only modest modification to existing code, we will release all source code used in this work to ease reproducibility.

## Footnotes

[1] Global decorrelation coupled with a linear classifier is efficient as the two linear operations can be merged.

[2]http://vision.ucsd.edu/~pdollar/toolbox/doc/

[3]http://www.vision.caltech.edu/Image_Datasets/CaltechPedestrians/

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
