[Reviews · NeurIPS 2014]

Submitted by Assigned_Reviewer_4

The paper introduces an efficient feature transform of local decorrelation, which when combined with boosted (orthogonal) decision trees, considerably improves over the state-of-the-art on pedestrian detection. Overall, it is a clearly (and nicely) written paper with good analysis, enough details and solid experiments.

Pros:
- Very well written and executed paper
- Attention to detail
- Solid results
- Straight forward and intuitive method

Cons:
- Incremental from Hariharan et al. (not major, see later)
- If it claims ``Improved Detection'', as opposed to ``Improved Pedestrian Detection'', then I would have liked to see some more results on object detection or likewise.
- Some minor analysis and experiments I would live to see (see later)
- Some parts can be better explained (see later)
- Relevance to NIPS (see later)

Some comments and concerns:

Incremental nature -- Though it might seem that this paper is an incremental work, I would say that it has enough novelty to offer over [15]. Going from global to local decorrelation, and doing the right analysis for design decisions set it apart. Though the analysis are in line with [6], I don't see that as a negative; I rather welcome the analysis.

Detection vs. Pedestrian Detection -- I would suggest the title to be changed to reflect that this work is currently targeted towards Pedestrian detection. Otherwise, I would like to see some experiments (may be similar to [15], or some work which uses Channel features) to show the effectiveness on standard detection tasks. Just to be clear: I am not asking for more state-of-the-art results.

Experiments/analysis --
1. L289-92: How does downsampling the final set of feature impact the overall performance? Does it improve quantitative performance, or just improves efficiency at the cost some quantitative performance?
2. Table 2 (6. LDCF constant): This is a very interesting result for me. Using k DCT basis filters itself takes them from 17.3% to 14.2% MR. Can the authors perform more analysis on using these DCT basis? I'm thinking on the lines of Fig. 4 for DCT basis. Does varying the DCT basis or the size of the filters improves the performance further? What's the impact of using more/less tree depth and sampling interval? I think this analysis will be interesting, and will give a new perspective to using these DCT basis. However, I don't think higher performance from these constant filters undermines the impact/novelty of this paper.
3. L375-77 (Fig. 4c): I see that the performance is still improving with decreasing tree depth and sampling interval. Did the authors try to push it more? Does the performance plateau? It would be nice to see that. If it wasn't feasible to push further, it can be mentioned.
4. "Replacing oblique splits": I'm not in favor of stating that the current method can act as a replacement of oblique splits. To state that, I would like to see both are ultimately doing the same thing (may be by analyzing the hits/misses at test time etc.) But this is a personal preference.

Clarity --
1. Section 4 (Complexity): How are #patches=d? Use of \Sigma and \bar\Sigma etc. I think this part can be made clearer.
2. Section 5 (observations): I think it might be a good idea to ref. each observation to a section/sub-section to make the obvious connection.
3. (minor) L269: How many levels per octave in feature pyramid? Given that authors are releasing code, this might not be necessary.
4. Section 5 (Estimating \Sigma): Which large collection of images did the authors use? #images and #patches?
5. Caltech Results (L409-10): ACF-Caltech is the standard ACF paper trained on Caltech? ACF-Caltech+ is just using more training data in a more complex model? By the ``ability to effectively utilize ..", do the authors mean computational ability or something else? Some clarity in this part will help!
6. Fig. 5: Having citations with each result in legends might be a good idea. If authors are not citing all paper due to space limitation, partial citation list will also be helpful.

Relevance -- I think this paper might be slightly better suited for a vision venue as opposed to NIPS. But I definitely see merit and novelty in the paper, and its potential impact. I would argue for the paper!

Summary: I like the paper in general. Writing, experiments+results and attention to details are its strong points. I would like to see authors rebuttal on my comments, particularly on experiments/analysis.

Submitted by Assigned_Reviewer_16

This paper proposed a method for feature transform that removes the correlation in local neighborhood, which make the feature suitable for orthogonal decision tree. Results in a better performance and lower computation cost.
The proposed method is based on the observation that effective but expensive oblique splits in decision trees can be replaced
by orthogonal splits over locally decorrelated data. The idea is simple but principled, and can be performed efficiently. The method has been described clearly and with source code it will very easy to reproduce.

While There are many feature transform method in literary, the paper has clearly stated the difference and advantage of proposed method.

There are also serval other interesting observations are discussed in the paper. e.g. the standard ZCA-withen transform used by LDA is ineffective. This provide more insight to the problem and helps the understanding of the role Local deceleration.

Although the proposed idea is simple and build on previous work, the method for feature transform itself can be a important building block, and be used by many application, therefore it might have a potential to have large impact.

The power of proposed method will more convincing if evaluation can be done in multiple benchmarks, especially those challenging ones, e.g. PASCAL, ImageNet... It will also be very interesting to show how the local deceleration of feature can help on other vision tasks besides detection.
Summary: The proposed method local deceleration for feature transform is simple and principle, while more evaluation is necsaary to make the paper more convincing.

Submitted by Assigned_Reviewer_43

This paper presents a new improvement for cascade object detection model. The new idea is to apply "decorrelated features" to use "orthogonal decision trees". In contrast to orthogonal decision trees, oblique decision trees are less likely to overfit the training data, but much more expensive to train and use at testing stage. The authors propose to decorrelate training features in order to maximize the classification ability of orthogonal trees to match oblique decision trees. The feature decorelation is achieved by a LDA like approach. Experiments show that the proposed method outperform the state-of-the-art cascade detection model. However, the paper does not address the following question critical to object detection problem:
- what's the accuracy comparison between state-of-the-art part based models and cascade model.
- what's the speed difference between the proposed method and high-speed part-based model such as Hyun Oh Song's Sparselet Models for Efficient Multiclass Object Detection
- what's the accuracy comparison between applying decorrelated features to oblique decision trees and the proposed method
- Why a couple of hours difference in training time is critical? (table 1)
Summary: The paper presents a sound idea to improve the cascade object detection model. However, given the overall progress of object detection methods, the paper does not provide sufficient experiments to validate the advantage of the proposed method.
Author Feedback
Author rebuttal: We thank the reviewers for detailed and genuinely helpful reviews. We are encouraged that all reviewers saw the value of our proposed approach: R1: “idea is simple but principled, and can be performed efficiently”, R2: “straightforward and intuitive method”, R3: “a sound idea to improve the cascade object detection”. Both R1 and R2 see the potential impact of the approach: R2: “I definitely see merit and novelty in the paper, and its potential impact”, R1: “the method for feature transform itself can be an important building block … therefore it might have a potential to have large impact”. R1 states that “the method has been described clearly and with source code it will very easy to reproduce” and R2 adds “very well written and executed paper” and “solid results”.

The main concern shared by all reviews is that the paper focuses on pedestrian detection and not more general object detection: R1: “more convincing if evaluation can be done in multiple benchmarks … e.g. PASCAL, ImageNet”, R2: “I would have liked to see some more results on object detection or likewise”, R3: “given the overall progress of object detection methods, the paper does not provide sufficient experiments”. First, we openly acknowledge that while boosted object detectors achieve state-of-the results on quasi-rigid detection (faces, pedestrians, cars), these methods do not achieve top results for more general/articulated detection (PASCAL, ImageNet). Hence, without modification (e.g. adding parts), our proposed method would not work well on more general object detection. Interestingly, on pedestrian detection we easily outperform the top methods for general detection including variants of deformable part models (DPMs) [11,36,26] and also convolutional nets [33,37,25].

In short: we target quasi-rigid detection and in particular pedestrians. This is a critical unsolved problem with numerous applications (robotics, self-driving cars, mobile devices). We note that nearly *40* top-tier papers have been evaluated on the Caltech Pedestrian Dataset (see www.vision.caltech.edu/Image_Datasets/CaltechPedestrians/files/algorithms.pdf). Of these, much as in our own work, most focus exclusively on pedestrian detection. We substantially outperform all these methods, reducing false-positives *tenfold* on Caltech Pedestrians over previous state-of-the-art. R2 makes the excellent suggestion of changing the title from “Improved Detection” to “Improved Pedestrian Detection”. We agree and will do so if accepted. We hope that this along with corresponding changes to the text will help clarify the contribution of our work.

We now address remaining concerns.

R2: “paper might be slightly better suited for a vision venue as opposed to NIPS”. Our proposed transform improves decision trees given correlated features. Given that ensemble methods such as boosted trees and random forests were primarily developed and published in the machine learning community, we ultimately felt our paper was better suited for NIPS.

R2, 1: “How does downsampling the final set of feature impact the overall performance?”. No measurable impact on accuracy but improves speed. We will clarify.

R2, 2: “LDCF with DCT filters”: Indeed, this works quite well. Note that as discussed in last paragraph of page 6, for color channels our method basically discovers the DCT bases (although we use the filters in a different manner than the DCT transform). The main advantage of the learned filters is they can adopt to individual features channels (e.g. oriented gradients, see fig 3). As R2 notes, however, we show that just using tuned DCT-like filters for all channels already works reasonably well (Table 2).

R2, 3: “I see that the performance is still improving with decreasing tree depth and sampling interval.” Actually, it’s essentially plateaued (32.85% MR at interval=8,depth=5, 32.68% at int=4,dep=4, and 32.60% at int=4,dep=5). We will add details to text.

R2, 4: “I'm not in favor of stating that the current method can act as a replacement of oblique splits”. The experiments in section 4 show that using a single shared Sigma is superior to using separate Sigma per split (due to implicit regularization provided by using shared Sigma). Hence, after decorrelation, using oblique splits (which allows for a separate Sigma per split) actually slightly degrades performance. If accepted we will clarify.

R2, Additional comments on clarity: if accepted we will incorporate these helpful suggestions.

R3: “What's the accuracy comparison between state-of-the-art part based models and cascade model?” and “what's the speed difference between the proposed method and high-speed part-based model such as Hyun Oh Song's Sparselet Models?”. As discussed earlier, methods suited for general object detection typically are not well suited for quasi-rigid detection and vice-versa. Hence on PASCAL/ImageNet part-based methods would outperform our approach while for pedestrian detection our method does better (the top part-based method MT-DPM+Context [36] achieves 41% MR compared to 25% MR for LDCF, see figure 5). Speedwise, boosted cascades including the proposed approach are substantially faster than part-based methods.

R3: “What's the accuracy comparison between applying decorrelated features to oblique decision trees and the proposed method?” Oblique trees are unaffected by a rotation (decorrelation) of the features. After decorrelation, orthogonal trees outperforms oblique trees (see also response above to R2,4).

R3: “Why a couple of hours difference in training time is critical?”: The speedup is two orders of magnitude (5 hours versus 5 minutes on fairly small INRIA dataset). For larger datasets, including Caltech, this is critical. Moreover, note that oblique trees are slow at test time while orthogonal trees are extremely efficient and are routinely used for real-time detection.